# Microshape Measurement Method Using Speckle Interferometry Based on Phase Analysis

**Yasuhiko Arai**

Department of Mechanical Engineering, Faculty of Engineering Science, Kansai University, Yamate-cho 3-3-35, Japan; arai@kansai-u.ac.jp

**Abstract:** A method for the measurement of the shape of a fine structure beyond the diffraction limit based on speckle interferometry has been reported. In this paper, the mechanism for measuring the shape of the fine structure in speckle interferometry using scattered light as the illumination light is discussed. Furthermore, by analyzing the phase distribution of the scattered light from the surface of the measured object, this method can be used to measure the shapes of periodic structures and single silica microspheres beyond the diffraction limit.

**Keywords:** measurement beyond the diffraction limit; measurement mechanism; non-periodical fine structure measurement; speckle interferometry; detecting phase distribution

## 1. Introduction

When a rough surface is illuminated with a highly coherent light, scattered light is generated from the measured surface. The scattered lights interfere with each other and produce speckle, which is a grain interference fringe whose diameter depends on the size of the aperture of the objective in the optical system [1–3].

In interference measurements, speckle is sometimes treated as noise present in fringe images. Therefore, a reduction in the influence of speckle noise is important in speckle interference measurement technology [4,5]. Filtering technology has been studied and further developed to address speckle noise [5–7]. In addition, high-resolution measurements using techniques such as fringe scanning [3] have been widely employed in speckle interferometry. Several high-resolution measurement methods for deformation or displacement have been developed by analyzing the recorded phase distribution in the speckle [8–13]. Speckle is currently considered a significant phenomenon in the optical field. Furthermore, speckle interferometry has been used for three-dimensional (3D) shape measurements [14,15]. A 3D structure beyond the diffraction limit of the objective lens can be measured by using the scattered light as the illumination light and by detecting the phase of the light recorded in the speckle formed by the scattered light [15].

The principle of this method [14,15] can be validated by comparing its measurement results with those of other techniques, such as atomic force microscopy (AFM), using the grating of the periodic structure as the measurement object. In this case, because the same structure exists everywhere, the same situation of the same measured object can be measured [14–16]. However, if only the periodic structure is measured, misunderstandings and misconceptions can emerge if the speckle interference measurement technology is not well understood. It has been misunderstood that the proposed method can be used only for measuring the periodic structure. In addition, a misconception emerged that this method has the same principle as that of scatterometry [17–20], which is used to measure periodic structures.

To clarify such misunderstandings and misconceptions, this study demonstrates that the proposed measurement method can measure three-dimensional structures that exist randomly in space. Furthermore, the same measured object at the same sample position was measured under different conditions of the diffraction limit by installing an aperture

in front of the objective lens. Accordingly, the results were analyzed based on whether the diffraction limit exceeded. As a result of comparing the results under the two situations, it was confirmed that the proposed method can measure beyond the diffraction limit.

Furthermore, this paper explains the mechanism of the measurement process based on the principle of this measurement method using scattered light as illumination.

This paper provides an understanding of the processing of the method, which makes it possible to perform three-dimensional measurements by analyzing only the phase change of the scattered light from the object without using the higher-order diffracted light based on conventional imaging theory [21].

## 2. Method

In a conventional speckle interferometer [1], the laser beam was divided by half mirrors. One of the obtained beams was irradiated onto the measurement object with a rough surface. The other was irradiated onto a rough reference surface to produce the reference light. An interference is generated between the scattered lights from the rough surfaces. The interference fringes, displayed on the image sensor, are captured through lens [1,11–13].

When such an optical system is used, the sampled speckle pattern has a bias component and original signal component in the low-frequency region in the intensity distribution. This phenomenon is the same as the interference fringe intensity distribution for a general interferometer because speckle is also an interference phenomenon. Therefore, an image in which the significant signal component in the frequency domain superimposes on the bias is obtained [6]. Such an image with a high resolution can be processed and a high phase component can be extracted using temporal fringe scanning technique [1–3].

However, this analysis technique cannot be used for dynamic analysis because three sheets of speckle patterns are required. In addition, because each speckle before and after the deformation must overlap in the speckle interference measurement [1,22,23], the relationship between the deformation and speckle size poses a problem in large-deformation measurements when the three speckle patterns are captured [24].

However, when a plane wave is used as the reference beam and the wave plane is angled with respect to the object beam [5], a carrier signal can be provided in the speckle. By providing such a carrier signal and performing a Fourier transform on the captured speckle pattern, the bias and signal components can be separated in the frequency domain [6,7,23–26]. The bias component can be removed by extracting only the signal components in this process. Thus, the phase component can be obtained using only two speckle patterns before and after deformation [22,23].

Furthermore, the phase distribution of the specklegram between images was analyzed using the basic speckle interferometry [5]. The noise in the low-frequency region of these images was removed by a filter using the Fourier transform.

However, when lasers were used, speckles in images were historically treated as noise components, and methods to reduce speckle noise have been extensively studied. In the analysis method used in this study, the noise components in the speckle are reduced by a two-step filtering technique to ensure that they do not affect the analysis. Through this processing [7], phase analysis is achieved without affecting the signal components.

Using such filtering technologies, the phase information in speckle interferometry can be extracted smoothly by a camera and computer technology, such as electronic speckle pattern interferometry (ESPI) [1,2]. Deformation measurements with a resolution of several tens of nanometers were conducted using such technologies [22,23].

On the contrary, in terms of the fine structure used in this study, it is believed that it is impossible to obtain the final shape unless filtering is performed. In this study, as described above, processing using filtering technology for noise components in two different frequency bands is performed in two steps [14–16]. Therefore, it is possible to observe the fine structure smoothly.

The cross-section of the measured object is shown in Figure 1. In Figure 1a, the phase cross-section of the speckle pattern is shown when the measured object is set to the original position of the speckle interferometer. In Figure 1b, the phase cross-section of the speckle pattern is set at the time when the measured object is shifted by a small amount ($\delta x$). When a specklegram is calculated between these two speckle patterns, the phase change at each position can be analyzed with high resolution using speckle interferometry. Assuming that the phase distribution related to the shape of the measured object is f($x$), as shown in Figure 1a, the phase change caused by the lateral shift ($\delta x$) at Pa in the original position is the difference between Pa and Pb, i.e., f($x$) − f($x + \delta x$). If this phase change is divided by the shift ($\delta x$), the pseudo-differential value ($\partial f / \partial x$) of the phase distribution f($x$) related to the shape can be obtained, as shown in Figure 1. By integrating this differential value with respect to the shifted coordinates, the phase distribution f($x$) related to the shape of the original object can be obtained.

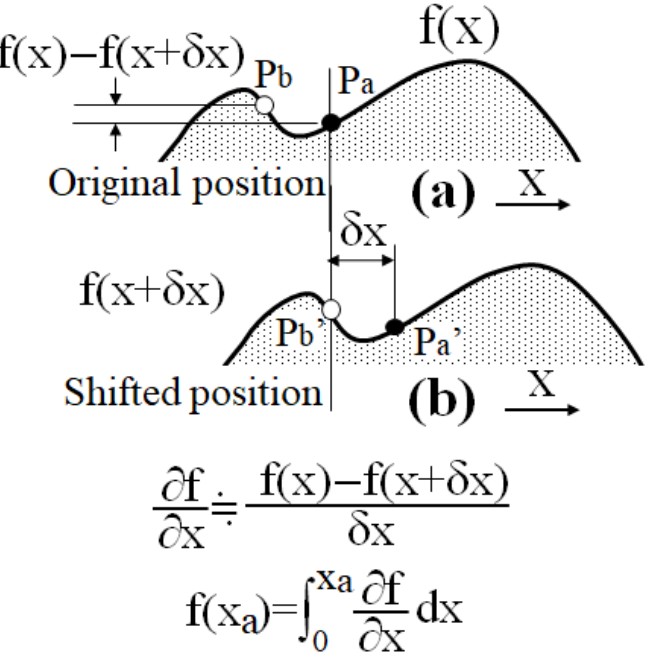

**Figure 1.** Principal of analyzing method: (**a**) Section of measured object at original position; (**b**) Section of measured object at shifted position.

As mentioned earlier, the lateral shift of the measured object is believed to be an "advantageous deformation for the speckle deformation measurement" for measurements of 3D shapes based on speckle interferometry. In addition, a piezo element was used in the initial experiment set for achieving lateral shift [14]. However, as the shift is several tens of nanometers, although the speckle pattern data collected without horizontal shifting are virtually shifted in the computer, it was reported that the result of the virtual shift produces the same effect as the result of the shift using the piezo element [15]. Therefore, in this analysis, only one speckle pattern was captured instead of collecting two speckle patterns before and after the horizontal shift. A second speckle pattern is produced as image data by virtually shifting in the memory of the computer [16].

Thus, the 3D shape can be measured using only one speckle pattern without any influence, such as mechanical vibration and fluctuation in air, from the experimental environment. In the 3D shape measurement, when a plane wave is used as the illumination light in the speckle interferometer, the scattered light from the measured object surface is largely affected by diffraction when the structure of the measurement surface becomes finer. Thus, the distribution of the diffracted light increases. When the higher-order diffracted light cannot pass through the lens area, the amount of scattered light passing through the lens from the surface of the measured object decreases. However, the decrease in the

amount of scattered light poses a problem in this method, based on the phase analysis of scattered light. In such a case, it is advantageous to use scattered light with many light direction vectors as illumination light to obtain more scattered light through the lens. Accordingly, scattered light was used as the illumination light at the time of measurement in this study.

The speckle interferometer used in this experiment is shown in Figure 2. In this optical system, scattered light generated after passing through the ground glass was used as the illumination light. The reflected scattered light from the surface of the object was collected by an objective lens (product number: M Plan Apo200×, NA: 0.62, magnification: 200×, Mitutoyo Co. Ltd.: Kawasaki, Japan). The scattered light, including the phase information, reaches the image sensor (pixel size: 1.6 μm, number of pixels: 1024 × 1024) through the pinhole, as shown in Figure 2.

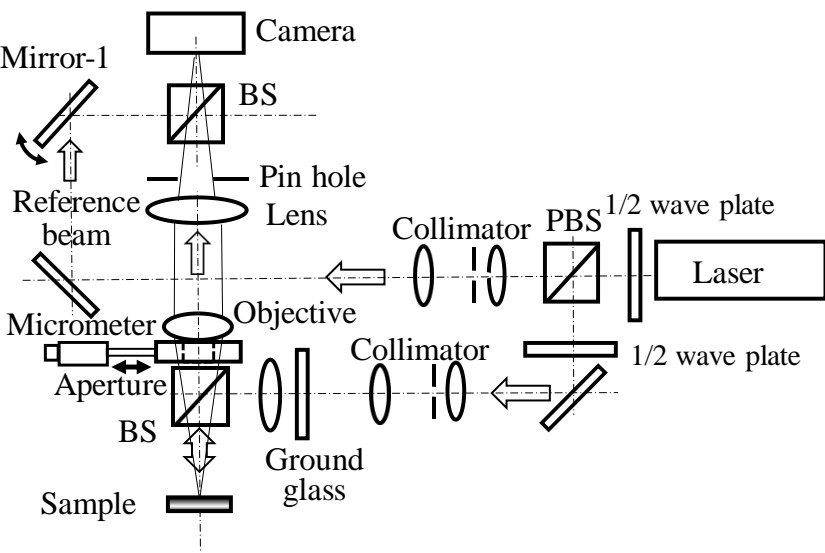

**Figure 2.** Optical system.

The beam emitted from the same laser (wavelength: 532 nm, 150 mW) is split by a polarizing beam splitter, collimated by a collimator, and used as the reference light. The camera captures the speckle pattern originating from the interference between the scattered light from the measured object and the reference light. The carrier fringes in the speckle are generated with the object light from the object surface and reference beam by setting the angle of mirror-1 to remove the bias component of the speckle pattern. The frequency of the carrier fringes is set considering the speckle diameter, transmission characteristics of the optical system, and camera characteristics shown in a previous report [24].

Furthermore, in this optical system, an aperture with a diameter of 8 mm on the rail and a thickness of 0.5 mm was installed in front of the objective lens using a micrometer. The NA of the lens is the original value of 0.62, when the aperture is not inserted. When the aperture was inserted, the aperture diameter of the lens changed from 13 to 8 mm, resulting in an NA of 0.29. Using such an aperture, the diffraction limit of the optical system can be changed without involving the measured object or the construction of the optical system [16]. In other words, the optical system can be used under two measurement conditions, except for the introduction of apertures. Therefore, an object on the order of several hundred nanometers can be observed by exceeding and not exceeding the diffraction limit. In this optical system, this can be achieved by changing the diffraction limit at the same measurement position of the same camera. The measurement results under different conditions are compared.

It is usually challenging to measure the microstructure (on the order of hundreds of nanometers) of the measured object at the same location using different measuring instruments. Therefore, in previous studies [14–16], the principle was validated by over-

coming this problem using a diffraction grating with a periodic structure by comparing it with other measurement techniques, such as AFM. This study verifies that this method can measure not only the shapes of periodic structures but also the shapes of randomly distributed 3D objects.

In this study, the shape of the microstructure randomly distributed in 3D was measured. The measured objects were silica spheres, whose shapes were not periodic. In addition, the diameter of the spheres was standardized. Here, because it is challenging to measure the same position of a random object using different measuring instruments, the measurement method was validated by comparing the measured diameters of the silica sphere, which has a standardized diameter under different measurement conditions. The diameter of the silica sphere was measured under different conditions of the diffraction limit without changing the position of the object.

The silica spheres used in this experiment were KMP-590 and X-52-854 manufactured by Shin-Etsu Chemical (Tokyo, Japan). The former is a sphere with a standard diameter of 2 μm (diameter distribution: 1–4 μm), while the latter is a sphere with a standard diameter of 700 nm (diameter distribution: 0.2–5 μm). As shown in Figure 3, the powdered sample microsphere was fixed to a 10 mm × 5 mm test piece with an adhesive to prepare the sample for measurement. In addition, the surface of the object was coated with platinum with a thickness of several nanometers by sputtering.

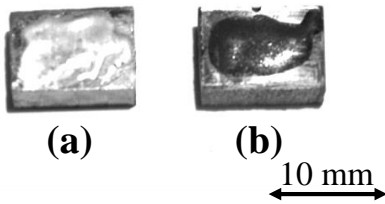

**Figure 3.** Test pieces with the microspheres (Silica sphere), (**a**) Diameter: 2.0 μm, (**b**) Diameter: 700 nm.

The diameter distribution of the measured object was observed using scanning electron microscopy (SEM) before performing the measurements. The results are shown in Figure 4.

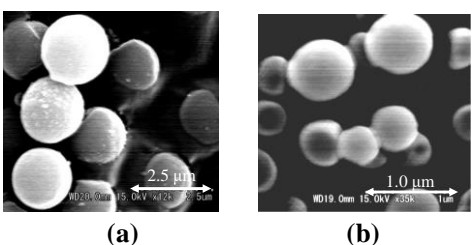

**Figure 4.** SEM image of measured objects: (**a**) Diameter: 2.0 μm (KMP-590); (**b**) Diameter: 700 nm (X-52-854).

The SEM image shows that the average diameter of the KMP-590 spheres was approximately 2 μm, while that of X-52-854 was approximately 700 nm. The measured diameter of the sphere was consistent with that in the official product catalog. Although the measured object is approximately spherical, it is not necessarily an ideal sphere; some ellipsoids as distorted spheres are observed.

## 3. Results

### 3.1. Shape Measurement of Randomly Distributed 3D Objects Not Exceeding the Diffraction Limit

During the observation of the measured objects with diameters of 2 μm using the optical system shown in Figure 2, a speckle pattern within a region with dimensions of approximately 20 μm × 20 μm was captured, as shown in Figure 5a.

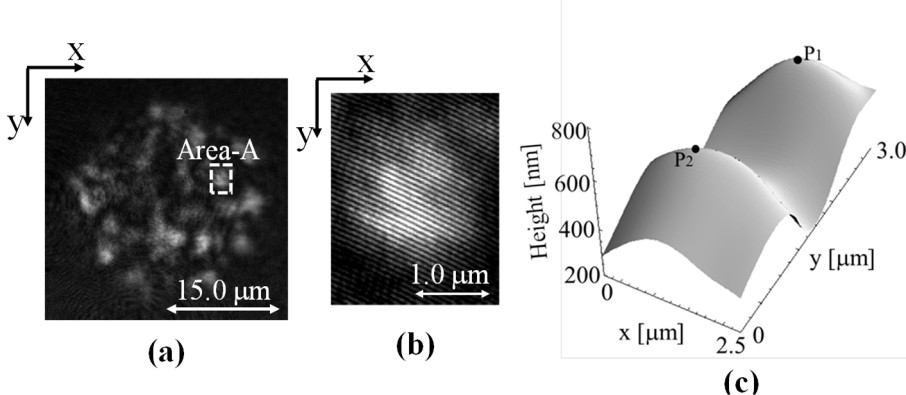

**Figure 5.** Speckle pattern and measured result: (**a**) Measurement area in the speckle interferogram; (**b**) Area-A; (**c**) Three-dimensional shape.

The bright parts were observed in the speckle pattern. In speckle interferometry, the measured objects are only the silica spheres distributed before and after the conjugate point of the image formation position of the lens. The field depth of the objective lens used in this study was ±692 nm because the NA was 0.62, and the laser wavelength was 532 nm. The value of the field depth of the objective lens was calculated using the formula $\lambda/(2NA^2)$, which was introduced by its manufacturer, i.e., only the silica spheres existing in the range of ±692 nm can be measured during the measurement of the randomly and spatially distributed silica spheres using this method. Therefore, the phase distribution in Area-A, which is a bright spot as a speckle, was measured because it was assumed to be the measured area. An enlarged view of Area-A is shown in Figure 5b. Some speckles were scattered, and carrier fringes were observed. Carrier fringes are used to remove the bias component of the speckle pattern.

The high-resolution phase distribution of Area-A (2.5 μm × 3 μm) of the speckle pattern shown in Figure 5a, which is the bright area of the speckle pattern, was calculated by the principle of this method as a three-dimensional shape map, as shown in Figure 5c.

In this operation, though this process has been discussed in the previous papers [14,15] in detail, the analyzing process is briefly introduced with the flowchart shown in Figure 6.

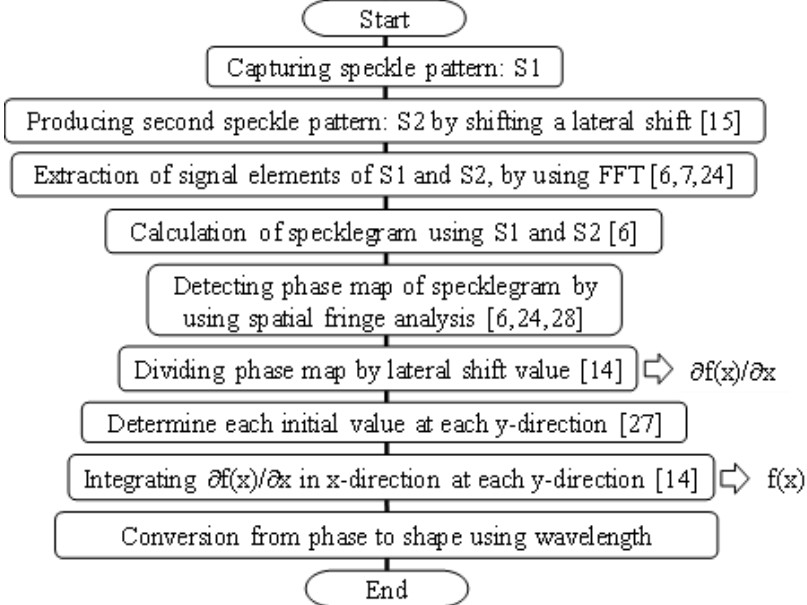

**Figure 6.** Flowchart of processing in Figures 5 and 7.

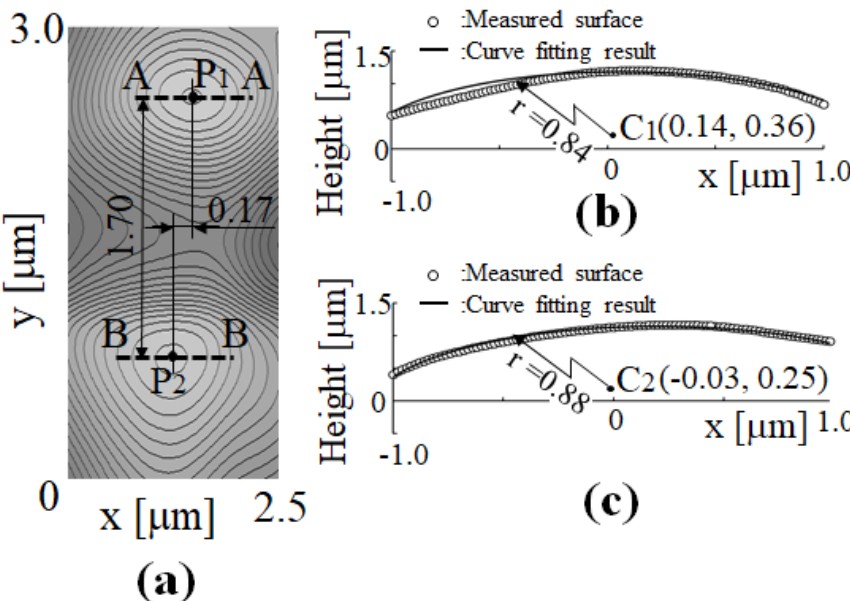

**Figure 7.** Measured results: (**a**) Contour of 3D shape; (**b**) Measured shape of section A-A; (**c**) Measured shape of section B-B.

In this process, first, one speckle pattern (S1) is captured. The acquired image is shifted by a lateral shift to generate the second speckle pattern (S2) [15]. From these speckle patterns, a process using the Fourier transform is performed to obtain the specklegram [6,7,24]. A high-resolution phase map corresponding to the derivative of the shape of the measurement object is extracted by using the spatial fringe analysis method under the idea of phase difference method [6,27]. The shape of the measurement object is then obtained by integrating the derivative distribution in the direction of lateral shift [14,15].

In the two-dimensional calculation of the shape, the current operation is only shifting in the $x$-direction, and there is no information in the $y$-direction. Therefore, the operation in the $y$-direction was performed by using the idea in previous paper [28]. In the $y$-direction operation, the initial value of the $x$-direction operation at the position away from the measurement object where the derivative distribution of the shape in the $y$-direction is considered not to exist can be assumed zero. Then, the results of integration in each $x$-direction from such positions were simply arranged as the two-dimensional result in the $y$-direction.

Two peaks (vertices), P1 and P2, with different heights, were observed in the 3D shape map in Figure 5c. Furthermore, Figure 7a presents the contour map of Figure 5c with a contour width of 50 nm. The difference in height was also observed when these measurement results were displayed on a contour map, as shown in Figure 7a.

Although the measured silica object includes some irregular spheres, most of the objects are approximately spherical, as shown in Figure 4. The vertex of the silica sphere can be easily estimated by observing the contour map shown in Figure 7a. Considering that the measured object is a sphere, it can be assumed that the horizontal cross-section, including the vertex on the contour map, is the cross-section of the sphere. Thus, the A–A and B–B cross-sections that include P1 and P2 can be set parallel to each $x$-coordinate. Therefore, the cross-section of the sphere can be easily obtained. Furthermore, as shown in Figure 4, a 3D estimation method based on the exact calculation of the diameter was not selected because the width of the diameter distribution is between 1 and 4 μm and silica objects are not necessarily ideal spheres. The measurement results by two-dimensional curve (circle) fitting on the cross-section using the least squares method instead of the 3D fitting method were evaluated to estimate the radius and center position of the object.

Peak P1 in section A–A is higher than peak P2 in section B–B, as shown in Figure 7a. The centers of the cross-sections A–A and B–B can be represented by C1 and C2, respec-

tively, as shown in Figure 7b,c. In the curve fitting, the radii of the silica spheres were approximately 840 and 880 nm in the A–A and B–B sections, respectively. Therefore, the diameters of the silica spheres centered on C1 and C2 were assumed to be 1.68 μm and 1.76 μm, respectively. Although the diameter of each silica sphere was slightly smaller than the standard diameter shown in the official product catalog, the measurement results were within the diameter distribution of the sample.

The 3D distance between C1 and C2 in the *y*- and *z*-directions by curve fitting of the A–A and B–B cross-sections was calculated using the 3D coordinates of the measurement results based on only *x*-direction information.

As shown in Figure 7a, the distances in the *x*- and *y*-directions were 170 nm and 1.7 μm, respectively. As shown in Figure 7b,c, the height difference between the two centers was 110 nm. Therefore, the 3D distance between C1 and C2 was approximately 1.71 μm. In addition, the center-to-center distance estimated using the sum of the radii of the two silica spheres was 1.72 μm. As the distance between the centers estimated by the results was 1.71 μm, the two silica spheres were considered to be in contact with each other. Therefore, it is possible to observe the shape of microstructures in contact with each other, if the diffraction limit was not exceeded.

In this study, silica spheres distributed in different regions, as shown in Figure 5a, were measured in other bright areas, similar to Area-A shown in Figure 5a.

As described above, this method can be used to observe the silica sphere existing at the imaging position of the lens at a field depth of ±692 nm. The measurement results for each bright speckle pattern area are shown in Figure 8.

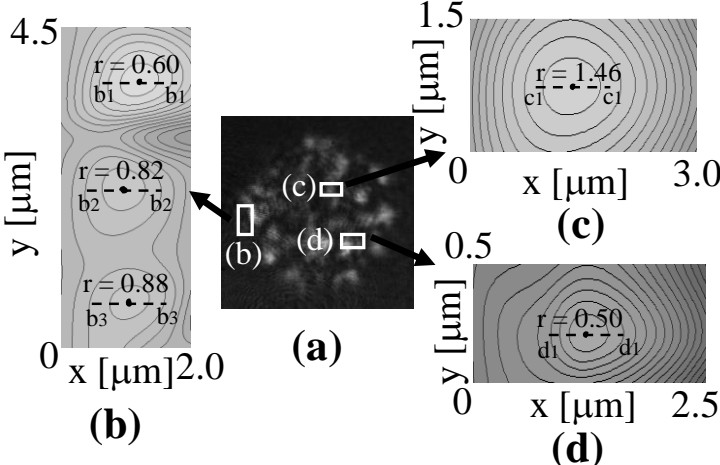

**Figure 8.** Measured results: (**a**) Measurement area in the speckle interferogram (same image of Figure 5a); (**b**) Measured shape in area; (**c**) Measured shape in area; (**d**) Measured shape in area.

As shown in Figure 8b, three silica spheres were connected. The radius of each sphere was calculated by curve fitting in the same manner as in Figure 7c. The diameters of the three spheres were approximately 1.3–1.8 μm. In addition, the estimation of the radius of the three silica spheres led to the assumption that the spheres are in contact with each other, similar to Figure 7a.

Furthermore, a single silica sphere can be measured, as shown in Figure 8c,d. The measurement results of a silica sphere with a large diameter of approximately 3 μm are shown in Figure 8c. Figure 8d shows the measurement results of a silica sphere with a small diameter of approximately 990 nm.

Therefore, this method can be used even when randomly distributed microstructures are in contact with each other.

Furthermore, the speckle pattern shown in Figure 5a has bright areas in addition to the bright regions shown in Figures 7 and 8. The diameters of 24 microspheres in these

18 bright regions were measured by this method. As a result, the average value of the diameter was 1.85 μm and the standard deviation was 0.633 μm.

The t-distribution [29] was used to a statistical test of the difference between the average values of the catalog data (2 μm) and the measured results. The t-distribution value can be calculated as $(2.0–1.85)/(0.633/24^{0.5}) = 1.16$, since the degrees of freedom are 23. The level of significance is set as 10%. From the t-distribution table, since it can be referenced that t(23,10%) is 1.714 (>1.16), it can be confirmed that the measured average of this method is clearly the same as the average value in the catalog data.

From the above results, it can be concluded that this method can measure the diameter of microspheres, provided that the diffraction limit is not exceeded.

### 3.2. Shape Measurement of Objects with Random 3D Distribution beyond the Diffraction Limit

When the measured object does not exceed the diffraction limit, the proposed method can be used to measure the shape of a sphere, even if the spheres are in contact, as shown in the previous section.

When the size of the measured object is smaller than the diffraction limit, the possibility of measuring the shape of a distributed object in 3D is investigated using a silica sphere with a standard diameter of 700 nm (diameter distribution: 0.2–5 μm).

The diameter of the silica sphere was approximately 700 nm, according to the SEM image in Figure 4b. For the optical system shown in Figure 2, the diffraction limits were 523 and 1119 nm in the cases without and with the aperture, respectively. Therefore, the shape measurement can be performed under two different measurement conditions of the diffraction limit by measuring the same sample. The results obtained with the same sample can be then compared.

The speckle pattern obtained without the aperture is shown in Figure 9a, in which Area-B is defined as the bright region of the speckle pattern. The area surrounded by the broken line in the enlarged view was analyzed using this method. Carrier fringes can be observed diagonally in an enlarged view. The shape measurement results are presented in Figure 9c. The contour map of Figure 9c is shown in Figure 10a.

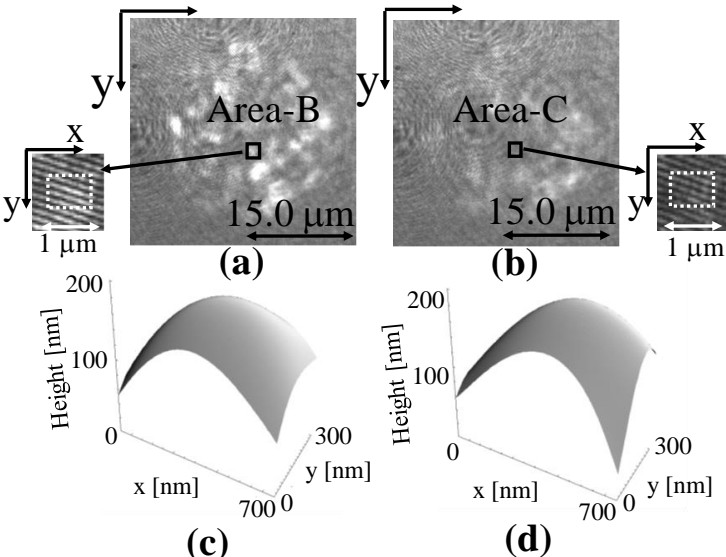

**Figure 9.** Measured 3D shape of 700 nm sphere at the same position: (**a**) Speckle pattern without aperture (diffraction limit: 523 nm); (**b**) Speckle pattern with aperture(diffraction limit: 1119 nm); (**c**) Phase map of measured 3D shape without aperture in Area-B; (**d**) Phase map of measured 3D shape with aperture in Area-C.

The cross-sectional shape (C-C), which is parallel to the *x*-direction, including vertex P3 in Figure 10a, is shown in Figure 10b. The radius and center position of the silica sphere

calculated for this cross-section by the curve fitting process shown in Figure 7 were 390 nm and C3 (10, −300), respectively. Hence, the silica object can be defined as a sphere with a diameter of 780 nm.

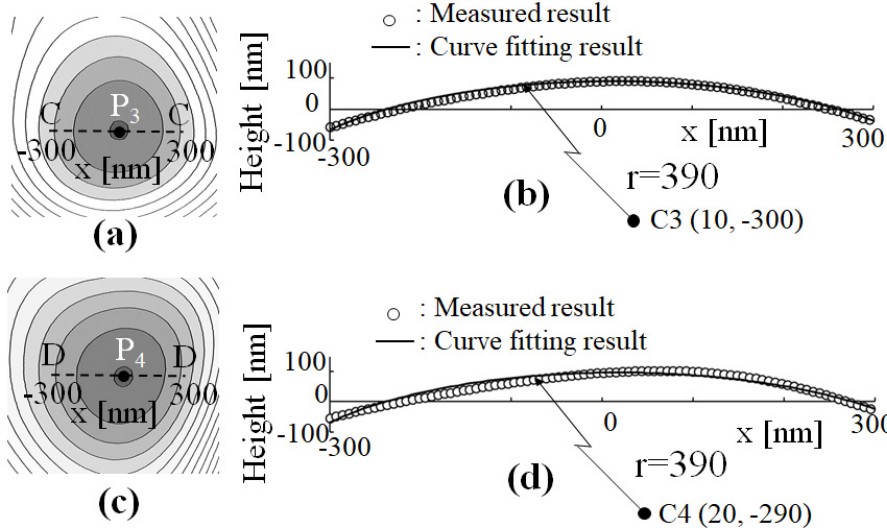

**Figure 10.** Measured 3D shape of 700 nm sphere at the same position: (**a**) Contour of measured 3D shape without aperture in Area-B; (**b**) Measured 3D shape without aperture in Area-B; (**c**) Contour of measured 3D shape with aperture in Area-C; (**d**) Measured 3D shape with aperture in Area-C.

This calculation is performed by the flowchart shown in Figure 6 as well as the results shown in Figure 7. These results also show that this method can be used to perform measurements if the size of the measured object does not exceed the diffraction limit. Furthermore, the diffraction limit was changed to 1119 nm by inserting the aperture in front of the objective lens of the optical system. The sphere in Area-C shown in Figure 9b was measured using an optical system that can measure the same sample at the same location after changing the diffraction limit conditions. Area-C in Figure 9b is at the same location as Area-B in Figure 9a. However, the speckle pattern in Figure 9b is generally darker and more ambiguous than that shown in Figure 9a. The 3D shape obtained by analyzing Area-C is shown in Figure 9d. Figure 10c shows a contour map of the results shown in Figure 9d. The results shown in Figure 9c,d are compared.

In the same manner as that for the results (C-C) in Figure 10b, the shape of the cross-section (D–D) centered on P4 in Figure 10c was estimated by curve fitting to determine the radius. A radius of 390 nm and center position C4 (20, −290) were estimated by curve fitting, as shown in Figure 10d. The results are approximately in agreement with the results shown in Figure 10b. This confirms the effectiveness of the proposed method, which can be used to measure the shape of a fine structure for a randomly distributed even if the diffraction limit is exceeded.

In addition, the diameters of the microspheres at six areas in the speckle pattern shown in Figure 9b were measured, although the insertion of the aperture reduced the number of bright regions in the speckle pattern and made data collection difficult. The average value of diameters was 1.09 μm, and the standard deviation was 776 nm. The statistical test using *t*-distribution [29] was also performed to compare the measured average value with the catalog data (700 nm). In calculation, $t = (1091–700)/(776/6^{0.5}) = 1.236$ was obtained. The level of significance is set as 10%. From *t*-distribution table, it can be checked as *t*(5,10%) is 2.015. Since *t*(5,10%) = 2.015 > 1.236, it can be confirmed that the average of the measured diameters is equivalent to the average value in the catalog data.

From the above results, it can be also concluded that this method can measure the diameter of microspheres, provided beyond the diffraction limit.

## 4. Discussion

The above discussion confirms the possibility of measuring the shape of microstructures beyond the diffraction limit of the objective lens by using scattered light as illumination light. Next, the logical mechanism of the measurement process is analyzed.

In an optical system based on the concept of perfect imaging [21], scattered lights are emitted from a point Pc on the object surface when illumination light is applied to a rough object surface, as shown in Figure 11.

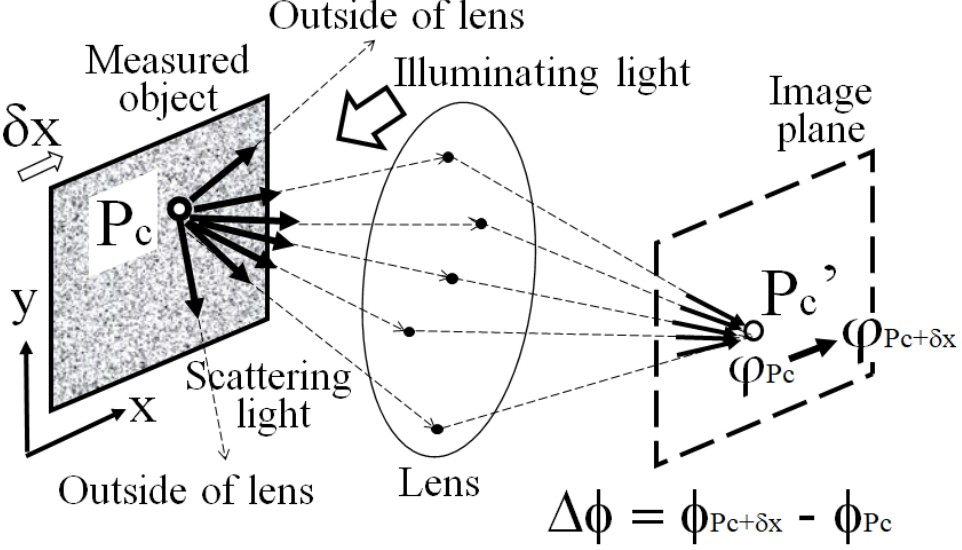

**Figure 11.** Model of measurement process.

Some of the reflected scattered lights converge to Pc' on the image plane of the lens after passing through random positions in the lens aperture. Therefore, when the measured object is set farther than the focal length of the lens on the optical axis, the reflected light from the point (Pc) on the surface of the measured object passes through the lens and converges at the corresponding point (Pc') of the surface of the measured object. The light emitted from one point on the object surface is focused on the conjugate point considering a perfect optical system. In this case, if the higher-order diffracted light from the measurement object passes through the lens and reaches the image plane, a clear focused image is observed, in accordance with Abbe's imaging theory [21].

However, when a fine structure beyond the diffraction limit of the objective lens is observed, higher-order diffracted light extends to the outside of the aperture of the lens and cannot pass through the lens, as shown in Figure 11. Therefore, such light cannot reach the image plane. Thus, a focused image could not be captured. According to traditional image theory [21], this imaging process can be attributed to an analysis method based on the formation of a light intensity distribution called an image.

When the measurement object is illuminated using scattered light, it is considered that a part of the scattered light from the measured object can pass through the lens aperture, even if the measurement object has a fine structure. As shown by the broken line after scattering at Pc in Figure 11, coherent light passing through various paths interfere with each other. Such scattered light information from each point on the measurement surface is aligned on the conjugate point of the measured surface under the concept of perfect optics.

In other words, because the speckle pattern reaches the image sensor through the lens under the perfect optical system, it is assumed that two-dimensional phase information (i.e., shape information) that has a one-to-one correspondence with the position of the measured surface is spatially aligned in a regular manner on the image sensor. Because this method uses phase information, it does not require higher-order diffracted lights for shape measurement analysis, as in the conventional imaging theory based on light

intensity distribution. If a light with phase information reaches each pixel of the camera through the lens, the phase of the light on each pixel is analyzed with a high resolution, and two-dimensional shape information can be detected on the image sensor using speckle interferometry. If the measured object is laterally shifted by $\delta x$ and the phase of the speckle from the neighbor point (Pc + $\delta x$) of the measured surface can be accurately detected, the two-phase information ($\phi$Pc before the lateral shift and $\phi$Pc + $\delta x$ after the lateral shift) at Pc' can be sequentially detected according to the principle of the 3D shape measurement.

The difference ($\Delta\phi$) of these phases is equal to the phase value at Pc' of the specklegram. This specklegram can be calculated between the speckle patterns captured before and after the lateral shift. The derivative value of the shape at Pc can be obtained by dividing the difference in phase by the lateral shift $\delta x$. By this operation, the two-dimensional distribution of the derivative of the phase regarding the shape of the measured object can be constructed by the computer. This information has a spatial one-to-one correspondence with the surface of the measured object. The original phase distribution f($x$) can be reconstructed by integrating the derivative value in the $x$-direction.

As stated above, it is considered that the 3D shape of the fine structure exceeding the diffraction limit can be analyzed by this method without the influence of diffraction and without using the concept of image formation, which is based primarily on intensity distribution. In addition, the scattered light with multiple ray vectors in various directions is indispensable as illumination light to perform this operation.

## 5. Conclusions

Although it has been reported that the shapes of periodic microstructures can be measured using this method, its usefulness for measuring randomly distributed 3D microobjects has not been demonstrated. In this paper, the feasibility of measuring randomly distributed 3D objects that exceed the diffraction limit with only one speckle pattern is discussed.

Verification experiments were performed using silica microspheres with standardized diameters. The possibility of the shape measurement of the fine objects randomly distributed in 3D was discussed using only a single speckle pattern, even if the size of the object exceeds the diffraction limit. As results of the test based on the statistical processing between catalog data and measured results, it is confirmed that the method can be applied to measure 3D structures randomly distributed in space.

In addition, the hypothesis of the measurement physical mechanism that the combination of the concept of perfect optics and the high-resolution phase detection technique using speckle interferometry with scattered light as illumination light enables the 3D shape measurement of microstructures beyond the diffraction limit was discussed. To verify this hypothesis, it is thought that further in-depth investigation and experimental results in wide science fields will be required.

In a future study, the limit of the measurement resolution of this method using scattered light will be investigated. The limit of the measurement resolution of this method depends on many parameters, including the lateral shift, which has already been discussed [15]; characteristics of the optical system, particularly the resolution of the lens; imaging resolution depending on the number of pixels and pixel size of the camera; speckle size; optical characteristics of the scattered light; and number of bits during the analysis by the computer. Furthermore, the capabilities of the arithmetic system and the influence of the measurement environment must be considered. These issues will be investigated in detail using experiments and computer simulations.

**Funding:** This work was supported by JSPS KAKENHI grant Number 20H02165.

**Institutional Review Board Statement:** Not applicable.

**Informed Consent Statement:** Not applicable.

**Data Availability Statement:** Not applicable.

**Acknowledgments:** Y. Arai thanks S. Yokozeki in Jyouko Applied Optics Laboratory for help discussion of this work.

**Conflicts of Interest:** The author declares no conflict of interest.

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
