# Peer review of "Microshape Measurement Method Using Speckle Interferometry Based on Phase Analysis"

_photonics, doi:10.3390/photonics8040112_

Round 1

Reviewer 1 Report

This paper describes how to obtain the microshape of measured object by collecting only one speckle interferogram. The author uses the optical system to collect one speckle pattern, and the second speckle pattern is produced by virtually shifting in computer. By adding an aperture in front of the microscope objective, the diffraction limit can be changed. Through the measurement of objects of different sizes, it is proved that the method is useful even if the diffraction limit is exceeded. Though the method proposed is quite useful, I cannot recommend acceptance owing to some technique questions because of those problems listed below:

  1. From the formula in Figure 1, is obtained after integral calculation. In my opinion, this value does not correspond to the topography. (line 122)
  2. In the experiment, the reference object is a mirror surface and the measured object is a rough surface. How to ensure the high fringe contrast?
  3. Will the actual size of the test object be affected when it is coated with platinum? (line 194)
  4. It is better to give the calculation and derivation formula of the depth of field for readers to understand. (line 220)
  5. How was Figure 5.(c) and Figure 6.(a) obtained?
  6. How are the positions of A and B located on the left side of part A and part B selected? (line 243)
  7. In Figure 6, how is the position of C and the radii of the silica spheres obtained? There is only a large diameter distribution range and no accurate values. How to judge the accuracy of these two radii?
  8. The English language and style require extensive editing. For example:

line 28: The following expression “ in the present, speckle is considered” should   be modified to “In the present, speckle is considered as”.

line 36-39: This sentence should be rewritten.

line 28: The following expression “ divided” should be modified to “divided by”.

line 103: The following expression “ un-less” should be modified to “unless”.

line 145-146: This sentence should be rewritten.

line 206-207: This sentence should be rewritten, etc.

Reviewer 2 Report

The paper presents an interesting approach  for the measurement of the shape beyond the diffraction limit using speckle interferometry however, the authors have previously published similar work showing the technique for periodic structures and as such the extension to microsphere seems rather incremental.

The paper is also overly long and repetitive in places (for example section 2 describing the well-known and established phase analysis technique using FFT [ Takeda 1990] . It is also conversely, lacking in detail about the method used to evaluate the phase gradient [line 118 in manuscript] - the size of the digital shift used, and how phase differences between the two points due to the tilt fringes are taken into account in this process?

In methods [around line 196] it is not clear how the same microsphere was selected for both SEM and speckle interferometry measurements. Figure 3 needs labels for a) and b), some indication of scale, and what is shown i.e. labels overlaid, and the  caption improved - as it appears to actually show the 10x5mm test object not single spheres themselves. Figure 4 there is no indication of the scale.

In results section; Figure 5 a) caption actually shows the speckle interferogram rather than speckle pattern directly. And as above the method used to calculate the three-dimensional shape (shown in figure 5c, 6 etc) is not described.

Finally, the conclusion that the shape could be measured is not fully justified. How was the choice of sphere /region of interferogram selected for analysis? Does the conclusion still stand if the distribution of sizes is calculated for a large number of spheres. If a distribution was measured for the sample as a whole and this was compared to the SEM measured distribution the conclusion would be much firmer - and i feel that this is essential if this paper is too be accepted.

Round 2

Reviewer 1 Report

The author has made many changes to the article and the writing errors have been reduced, but the core algorithm is still not convincing. From my perspective, if the algorithm cannot be explained more reasonably, the authors should not publish this manuscript.

  1. The author quotes his own articles (references 14 and 16) for the explanation of the formula in Figure 1. Please explain how the and f(x) in references 14 and 16 are obtained? The author first uses f(x) to find , and then uses to find f(x). This calculation process uses an unknown quantity to obtain another unknown quantity, which is unreasonable.
  2. The two images before and after are moved a certain distance instead of continuously changing. How to perform continuous integration?
  3. The experimental results in the article are three-dimensional topography, how is the y-direction reflected?
  4. The difference in light intensity can be obtained from the two pictures. If you use the light intensity difference to find the topography, because the article is a speckle diagram instead of a fringe diagram, the maximum measured value will not exceed the wavelength. But in your experiment, both 2000nm and 700nm exceed the wavelength.
  5. A beam splitter and a diffractive device are placed between the objective lens and the measured object, so the working distance of the objective lens is very long and the magnification will be small. The magnification of 200* in the article is unreasonable.
  6. line 328: The unit is um..
  7. In Figure 10, P0 should be changed to Pc, and P0' should be changed to Pc'.

Reviewer 2 Report

The authors appear to have made good progress in responding to my previous comments. (numbering used is from authors response)

1) The issues surrounding the work being incremental is acceptable to me.

2) The authors have rewritten and improved the manuscript to reduce the length in places.

3) Detail has been added in response to my previous comments (the size of the digital shift used, and I can now see my mistake concerning the tilt fringes bias.

4),5),6),7),8),9) -all minor and fixed.

However I am not satisfied with the response to the main points

10) “And as above the method used to calculate the three-dimensional shape (shown in figure 5c, 6 etc) is not described”

I still cannot see how the author response has fixed this issue and as it is key to the paper feel that at least a summary or diagram showing the processing steps should be included. The reference in the authors response giving the lateral shift is a good start but figure 5c does not show a method!

11)-13) My previous comment “Finally, the conclusion that the shape could be measured is not fully justified. How was the choice of sphere /region of interferogram selected for analysis? Does the conclusion still stand if the distribution of sizes is calculated for a large number of spheres. If a distribution was measured for the sample as a whole and this was compared to the SEM measured distribution the conclusion would be much firmer - and I feel that this is essential if this paper is to be accepted.”

I understand that it was not possible to identify and measure a single sphere using both SEM and the authors technique. However my point was that it is possible to measure a set of spheres using the SEM and determine a size distribution, in fact the authors give this measurement at line 254 in the revised manuscript:

“The SEM image shows that the average diameter of the KMP-590 spheres was approximately 2 µm, while that of X-52-854 was approximately 700 nm. The measured diameter of the sphere was consistent with that in the official product catalog”.

My point was that the same should be done using the speckle technique, i.e. more than few spheres presented should be measured and the mean diameter of these compared to the SEM mean diameter and catalog data.

Without this the measurement are to me still not wholly convincing as it may be there is a large uncertainty on the measurements with the selected sphere(s) just happening to lay close to the mean – whilst others could be very different but we don’t know as the authors only show selected measurements. It appears in figure 7a), 8a) and b) that there are multiple other spheres that could be measured other than those highlighted! It seems to me this small effort would make the paper much stronger.
